# Preparation and Separation Properties of Electrospinning Modified Membrane with Ionic Liquid Terminating Polyimide/Polyvinylpyrrolidone@Polydopamine

**DOI:** 10.3390/membranes12020189

**Published:** 2022-02-05

**Authors:** Peng Qi, Hongge Jia, Shuangping Xu, Qingji Wang, Guiming Su, Guoxing Yang, Mingyu Zhang, Yanqing Qu, Fuying Pei

**Affiliations:** 1College of Chemistry and Chemical Engineering, Heilongjiang Provincial Key Laboratory of Polymeric Composition Material, Qiqihar University, Wenhua Street 42, Qiqihar 161006, China; 15846275779@163.com (P.Q.); xshp_1979_1999@163.com (S.X.); zhangmingyuno1@163.com (M.Z.); vipquyanqing@163.com (Y.Q.); pfy1639916997@163.com (F.P.); 2CNPC Research Institute of Safety & Environment Technology, Beijing 102206, China; wangqingji@petrochina.com.cn; 3Institute of Advanced Technology, Heilongjiang Academy of Sciences, No. 52 Renhe Street, Nangang District, Harbin 150009, China; suguim@163.com; 4Synthetic Resin Laboratory Daqing Petrochemical Research Center, Petrochemical Research Institute, No. 2 Chengxiang Road, Wolitun, Longfeng District, Daqing 163714, China; ygx459@petrochina.com.cn

**Keywords:** polyimide, membrane, electrospinning, ionic liquid, oil-water separation fiber, polydopamine, polyvinylpyrrolidone

## Abstract

In this paper, superhydrophilic polyimide (PI) membranes were prepared using the electrostatic spinning method, capped with a hydrophilic ionic liquid (IL), and blended with polyvinylpyrrolidone (PVP). Using this preparation, the surface of the fiber membranes was coated in polydopamine (PDA) by means of an in-growth method. Scanning electron micrographs showed prepared blend films can form continuous fibers, for whom the distributions of diameter and pore were uniform. Post-modification (carried out by adding hydrophilic substances), the ability of the membrane surface to adhere to water was also significantly improved. The water contact angle was reduced from 128.97 ± 3.86° in unmodified PI to 30.26 ± 2.16°. In addition, they displayed a good separation effect on emulsified oil/water mixtures. The membrane flux reached a maximum value of 290 L·m^−2^·h^−1^, with a maximum separation efficiency reached of more than 99%. After being recycled 10 times, the separation efficiency maintained a level exceeding 95%. The purpose of this study is to demonstrate the simplicity and efficiency of this experiment, thereby providing new ideas for the future application of membrane separation technology in wastewater treatment.

## 1. Introduction

With the rapid development of oil extraction, chemical metallurgy, the textile and food industries, humanity is producing a large amount of wastewater containing a large proportion of oil and its derivatives. The means to quickly and efficiently treat oily wastewater is a matter of no little importance in current research. Traditional methods of treating oily waste [1,2] include gravity sedimentation [3,4], air flotation [5,6], separation by centrifuge [7,8], ultrasonic techniques [9], and electrochemical method [10], amongst others. However, these methods have poor separation efficiencies and high costs, yet can still easily cause secondary pollution. Due to its simple preparation, high efficiency, and broad applicability, membrane separation technology [11] has gradually attracted attention. At present, this technology is primarily used in such applications as gas separation [12], liquid separation, and ion-exchange membranes [13]. With the advancement of work on the separation membrane, this technology has now become the most effective method for treating oily wastewater [14,15].

This method is not only efficient and fast, but straightforward to use. Thus, it can be applied on an industrial scale while generating minimal secondary pollution [16]. To achieve effective separation, membranes need to have a standard in three basic features: infiltration performance; membrane pore size; and internal pressure. The construction of super-hydrophilic/oleophobic polyimide matrix composite membranes can theoretically drive a deeper understanding of the nature of such membrane separators, and provide new ideas for industrial applications [17,18,19].

Inspired by the oil-blocking properties of fish scales, a large number of superhydrophilic/oleophobic materials have been developed for treating water by applying this oil-blocking removal mechanism. At present, the preparation of these materials can be categorized into one of the following three approaches: surface coating; chemical surface grafting [20]; and modification by co-blending. Each of these methods of membrane hydrophilic modification have their own advantages and disadvantages.

Through chemical grafting, functional elements or groups can be introduced onto the main or side chains of a polymer. Two polymers with different properties can be grafted together to produce a new structure possessing special features. During the early years of developing this technique, Akhtar S et al. [21] grafted the amphiphilic polymer 2-methacryloyloxyethyl phosphorylcholine (MPC) onto the surface of polyvinylidene fluoride (PVDF) and cellulose acetate (CA) microfiltration membranes by using plasma etching. In the case of graft copolymers, due to the ease with which they are manufactured and the presence of sufficient active sites for reaction processes, they can be fully grafted and polymerized in solution or as part of an emulsion. However, solubility restrictions mean they are disadvantaged by low molecular weight.

Work has also been carried out in terms of blending. For example, Du. Q et al. [22] prepared a new nanofibrous membrane (NFM) by co-combining polyethersulfone (PES) with sulphonated poly(ether ether ketone) (SPEEK) by electrostatic spinning. Test results showed the composite NFM’s membranes were capable of separating not only immiscible oil/water systems but also oil-in-water emulsions. Thus, blending is one of the most common methods of membrane modification for good reason. Compared to other methods, the modification step is generally carried out simultaneously with the film formation, meaning it is simple and not restrained by cumbersome post-treatment steps.

Surface coating is an established and simple method of improving the hydrophilic effects of membrane surfaces. It directly or indirectly coats or precipitates a hydrophilic material with a specific functional group to form a hydrophilic layer on the membrane surface, thus improving the hydrophilic effect of the membrane. One approach reported in the literature was the preparation of nanoparticles on the polyvinylidene fluoride surfaces of microfiltration membranes formed by the co-deposition of anthocyanin and (3-aminopropyl)triethoxysilane (APTES) [23]. This nanoparticle-modified fibrous membrane surface is superhydrophilic and, when submerged, superoleophobic, which enables efficient separation of various types of emulsified oil/water systems. It achieved an optimum water permeability flux of dichloroethane in water emulsion of 187 L·m^−2^·h^−1^ under self-gravity. In addition, the rejection rate of various oils in water emulsions was consistently above 99.5%.

For membranes to be effective when applied to oil-water separation, they require excellent surface wetting properties, separation performance, and strong mechanical properties. Electrospun polyimide (PI) nanofibers are recognized as a special engineering material due to their extraordinary thermal and chemical stability. Due to the versatility of their structure, they are widely used in oil-water separation [24]. In a review on polyimides currently used in oil–water separation [25,26], Baig et al. concluded membrane surface roughness porosity is influenced by the polymerization process, thus, it affects the separation of oil–water mixtures and emulsions [27]. However, due to its easy adhesion to materials, such as organic fibers and inorganic particles, PDA will generate negative charges on the fiber surface, resulting in electrostatic repulsion between the particles. This strong electrostatic repulsion allows fibers wrapped by PDA to display extremely high stability. The work presented here used the electrostatic spinning method to produce a porous membrane with a suitable pore size. The ionic liquid (IL) was applied to regulate the polyimide film’s surface wettability. To compliment this, the addition of polyvinylpyrrolidone (PVP) enhanced film formation, and improved its mechanical properties and surface hydrophilicity. Finally, surface roughness was further improved, while simultaneously regulating pore size, by self-polymerizing PDA on the fiber membrane surface. These significantly improved the membrane’s properties as intended while also enhancing emulsified oil separation efficiency. A large variety of hydrophilic substances can be introduced into the membrane through simple in-situ growth, which can improve the membrane surface wettability while ensuring separation efficiency, meaning this research can be quickly applied to industrial production.

## 2. Experimental Section

### 2.1. Materials

The following compounds were used in this study: diamines in the polyimide 4,4′-diaminodiphenyl ether (ODA, 98% pure, and Mw 200.24), and 4,4′-methylenedianiline (MDA, 99% pure, Mw 198.28); and anhydrides 3,3′,4,4′-biphenyltetracarboxylic dianhydride (BPDA, 97% pure, Mw 368.43), 3,3′,4,4′-benzophenonetetracarboxylic dianhydride (BTDA, 96% pure, Mw 322.23), 4,4′-(hexafluoroisopropylidene)diphthalic anhydride (6FDA, 99% pure, Mw 444.24), and pyromellitic dianhydride (PMDA, 99% pure, Mw 218.12). All of the aforementioned chemicals were obtained from Aladdin Biochemical Technology Co., Ltd. (Shanghai, China). Further, dopamine hydrochloride (DA, 98% pure), (3-aminopropyl)triethoxysilane (APTES, 99.5% pure), tris(hydroxymethyl)aminomethane, (99.5% pure), and polyvinyl pyrroliketone (PVP, average mol wt 10,000, Mw 1300000) were supplied by Inokai Technology Co., Ltd. (Beijing, China). 1-Carboxyethyl-3-methylimidazolium chloride ionic liquid (98% pure, Mw190.63) was sourced from Lanzhou Institute of Chemification, Chinese Academy of Sciences (Lanzhou, China). In addition, *N-methyl* pyrrolione (NMP) of 98% purity was purchased from Common Chemical Reagents Co., Ltd. (Tianjin, China).

### 2.2. Polymerization

Figure 1 shows the synthetic route for the ionic liquid-capped polyimide. ODA and 6FDA were taken as examples to describe the preparation process. Initially, ODA (10.50 mmol = 2.10 g) was dissolved in NMP solution (10.0 mL) and transferred to a three-neck flask. Then, 2.00 mmol (0.88 g) of 6FDA was added to the reaction in 5 batches at 15 min intervals. After stirring for 1 h, this was followed by the addition of a 0.50 mol/L NMP solution of 1-carboxyethyl-3-methylimidazolium ammonium chloride (2.0 mL, mass concentration 0.1%). Finally, the NMP solution of PVP (2.8 mL, mass concentration 0.1%) was added to the solution which was left to stir for 6 h. The spinning solution of ionic liquid-4,4′-diaminodiphenyl-4,4′-(hexafluoroisopropylidene)diphthalic anhydride (IL-ODA-6FDA/PVP 14.8 mL, mass concentration 26%) was prepared during this time.

### 2.3. Membrane Formation and Thermal Treatments

14.87 mL of IL-ODA-6FDA/PVP solution was taken up into a syringe affixed on the electrostatic spinning machine. The film was spun with the following process parameters: voltage of 16 kV; spin cure distance of 15 cm; jet speed 0.8 mL/min; and the collector jet rate receiver speed set at 200 r/min. The electrostatically spun films were subsequently placed in an oven and the temperature gradually raised from 80 to 280 °C (5 °C/min until reaching 280 °C for 8 h). After the residual solvent was completely removed, the materials were maintained at 280 °C for an additional hour. Upon cooling to room temperature, the film was cleaned by ultrasound in ethanol and water to remove surface impurities. By these means, the polyimide blend porous membrane was prepared.

### 2.4. Coating Film Preparation

Tris (0.90 mmol = 0.12 g), DA (1.00 mmol = 0.20 g), and APTES (4.50 mmol = 1.0 g) were dissolved in ethanol/deionized aqueous solution (*v*/*v* = 1:5, 120 mL, pH 8.5) to serve as the medium of coating impregnation. After the fibrous membranes were left in the solution at room temperature for 24 h, they were washed with deionized water and dried for a further 3 h. This process resulted in the final IL-ODA-6FDA/PVP@PDA fiber membranes.

### 2.5. Emulsion Oil Preparation

Four different oil-in-water emulsions were prepared for this experiment. Petroleum ether, *n*-hexane, toluene, and trichloromethane (all in quantities of 10 mL) were each added to separate batches of 90 mL water (containing 3 wt% Tween 80, 0.3 mL) and stirred for 30 min to produce milky emulsions. These could be left for up to one week and still ensure a uniform dispersion in water.

Upon successful synthesis, the membrane flux and separation efficiency of the fiber membranes were assessed. In this paper, Formula (1) was used to calculate the former property. The flux, F, refers to the volume of flow through the effective membrane area in a fixed period of time. Since this test was performed without additional pressure, the unit is L·m^−2^·h^−1^, while V, A, and ∆T refer to the flow volume, effective membrane area, and unit time, respectively.
(1)$$F=\frac{V}{A∆T}$$

Typically, the separation efficiency is used to judge the effective degree of separation of the subject material, with higher water purity (and by association, ability to remove oil and other contaminants) representing a stronger effectiveness. Here, this was calculated using Formula (2), where C_o_ and C_p_ represent the concentration of the oil-in-water emulsion before and after the separation, respectively, which were both measured by an ultraviolet spectrophotometer.
(2)$$R\left(\%\right)=\left(1-\frac{C_{p}}{C_{o}}\right)\times 100\%$$

### 2.6. Membrane Characterization

Fiber membranes were prepared for analysis by electrostatic spinning machines as per the abovementioned method (SS series, Beijing, China). A Fourier transform infrared spectrometer (PE, Waltham, Massachusetts) and X-ray diffraction instrumentation (ESCALAB250X1, Thermo, CA, USA) provided infrared spectra and X-ray photoelectronic spectra, respectively. Surface morphology was characterized using a scanning electron microscope (S-3400, Hitachi, Japan). Mechanical properties were analyzed with a film tensile testing machine at 25 °C, with the standard spline’s dimensions of length 50 mm and width 20 mm, and a test speed of 5.00 mm/min (XLW(PC)-500N, Sumspring, Jinan, China). Video contact angle meter was tested by assessing the wettability of film surfaces, and the standard spline’s length and width both at 20 mm (JY-82B, Chengde Dingsheng Test and Testing Co., Chengde, China). Size characterization of the filtrates nanoparticles was carried out using a nanometer particle size analyzer (ZS90, Malvern, UK). Absorbance of the emulsion before and after separation was assessed by UV spectrophotometer, using 5 mL for each sample (Lamda35, Cincinnati, OH, USA). Lastly, macroscopic observation of sub-macro emulsion separation was undertaken using an optical microscope, 5 mL for each sample (E200, Nikon, Japan).

## 3. Results and Discussion

### 3.1. SEM Observation of Polyimide Membranes

As shown in Figure 2, the diameters of the ODA-6FDA, IL-ODA-6FDA, and IL-ODA-6FDA/PVP fibers were in the range of 3.53–4.89 μm. The microscopic surface was relatively clean and smooth as observed under different magnifications. In addition, for the porosity in Table 1, we used the gravimetric method to estimate the porosity of the membrane by soaking the membrane in ethanol and measuring the weight change before and after. According to Figure 2 and Table 1, the spun fibers’ porosity and diameter increased when capped with IL and blended with PVP. This was due to the ionic liquid raising the solution’s internal charge density, the fibers tend to elongate more during electrospinning thus will form thinner fibers. The addition of PVP also improved the viscosity of the solution and internal banding of fibers. To conclude, from the process as detailed, the addition of IL improved the conductivity of the spinning solution [28,29]. The overall trend of the spun fiber diameters remains broader, with a more uniform diameter distribution across all fibers. After surface coating modification, affixation of self-agglomerated PDA nanoclusters to the porous membrane was observed, as shown in Figure 2d. The introduction of PDA did not change the fibers’ basic morphology. This is because PD itself can be deposited as a conformal film on almost all types and sizes of organic and inorganic surfaces through a simple dip coating process, which is also different from other surface modification methods. In most existing theories of PD formation, two compounds (dopamine-quinone and 5,6-dihydroxyindole (DHI)) are its key building blocks, albeit through a variety of proposed pathways. After polymerization for 24 h, the sample of IL-ODA-6FDA/PVP changed color from white to dark brown, which directly proves DA was oxidized and polymerized to form PDA [30,31]. According to APTES, it can be hydrolyzed under alkaline conditions, resulting in a product which contains a lot of hydrophilic groups. Self-polymerization by DA under similar conditions as this hydrolysis can be realized at the same time. Co-deposition of APTES and PDA on the membrane surface through hydrogen bonding and intermolecular force using amino groups generated by hydrolysis [32]. The addition of a silane coupling agent (in this instance, APTES) enhanced the in situ growth of fiber membranes composed of PDA on IL-ODA-6FDA/PVP. For these membranes, the addition of PDA increased their roughness and boosted their hydrophilicity and underwater stability. This resulted in the improvement of the IL-ODA-6FDA/PVP fibers’ surface wettability and the membranes’ oil/water separation, amongst other properties.

### 3.2. Characteristics of Modified Membranes

In order to verify whether the ionic liquid reacted with the polymer matrix, FTIR analysis was conducted to determine the fiber membranes’ chemical structure. In preparation, the matrix was soaked in and washed with methanol for 7 days to remove residual ionic liquid to prevent it from affecting the test results. As per Figure 3, the –NH absorption shock peak at 1664 cm^−1^ indicated the formation of PAA, confirming the presence of its imide group. Compared with PAA, other samples showed C=O and C–N bond stretching vibration peaks in the imide group at 1712 and 1367 cm^−1^, respectively, while the aforementioned –NH absorption peak was no longer observed. These prove the formation of ODA-6FDA. Additionally, the C–H stretching vibration of the imidazole cation and skeleton ring, and the vibration absorption peaks of the nearby methyl C–H bond structure could be discerned at 3150, 1554, 2790, and 879 cm^−1^, respectively. With the addition of PVP, in contrast to IL-ODA-6FDA, the infrared spectrum of this modified system generated peaks at 3150 and 1620 cm^−1^. The spectra of the IL-modified system shifted these peaks towards higher frequencies by 6.2 and 4.5 cm^−1^, respectively, with large deviations. It demonstrated a strong interactive force between the components and a greater extent of compatibility.

As indicated in Figure 4, F1s (687 eV), C1s (283 eV), N1s (397 eV), and O1s (531 eV) were clearly visible in the spectrum of the original ODA-6FDA. By contrast, Cl2p (198 eV) was also observed in the IL-ODA-6FDA, demonstrating that IL was grafted successfully into the ODA-6FDA chain segment. Two new characteristic peaks in IL-ODA-6FDA/PVP@PDA were observed, indicating Si2s (130 eV) and Si2p (101 eV), showing PDA and APTES were successfully coated onto IL-ODA-6FDA/PVP fibers after functionalization.

### 3.3. Mechanical Properties of Membranes

The mechanical properties of different kinds of polyimide porous membrane were studied (see Table 2). Elongation at point of breakage of the membranes modified with IL and PVP was 8.9% to 39.21%. Their elastic moduli ranged from 2271.23 to 3056.46 MPa which was higher than in the original PI film. It can be concluded the addition of IL and PVP has a great impact on the mechanical properties of PI. Meanwhile, elongation at point of breakage of IL-ODA-6FDA was 26.95%, higher than that of the unmodified membrane by more than a factor of three. Its elastic modulus reached 2271.23 MPa, which is 4.9 times higher than that of ODA-6FDA. IL-ODA-6FDA/PVP also had a significantly superior elastic modulus, elongation upon breakage, and physical properties in general. When PDA was compounded on the IL-ODA-6FDA/PVP membrane, the mechanical properties of the resultant membrane were not significantly different from those of the unmodified membranes.

The improvements to the membranes’ mechanical properties as a result of the additions can be explained by the ionic liquid behaving as a blocking agent, adjusting the molecular weight. This had an impact on the arrangement of molecular chains, causing the orientation of ODA-6FDA to change, resulting in the mixed matrix film having a higher elastic modulus compared with pure PI. On the other hand, PVP was introduced into the pure film as a dispersed phase, though the compatibility of polyimide and PVP was superior, enhancing its mechanical properties.

### 3.4. Wettability of Membrane

Generally speaking, membrane surface wettability is an important feature of oil/water separation performance. Here, contact angle θ tests were conducted to explore the wettability of the synthesized membranes. Experimental data showed the ODA-6FDA fiber membrane’s (θ) value to be in the range of 126–129°, which did not change significantly after 20 s, meaning this unmodified membrane was hydrophobic (Figure 5a). The IL-ODA-6FDA fiber membrane’s θ value decreased from 128.97° to 114.70° (Figure 5b) over the testing period, during which water droplets gradually penetrated through the membrane. From this, it was concluded the addition of an ionic liquid can affect the wettability of the membrane surface. Compared with the former two, the IL-ODA-6FDA/PVP fiber membrane displayed better surface wettability (Figure 5c). Its θ value likewise changed over the course of the testing period, but diminished from the lower initial 64.43° to 30.26° (Figure 5d). Finally, the ODA-6FDA membranes coated with PDA exhibited consistently excellent hydrophilicity. The decreasing trend in water contact angles in most variants indicated surface wettability of the membrane was significantly improved after the cumulative addition of ionic liquid, blending PVP, and coating PDA. Consequently, it was concluded the addition of hydrophilic groups decreased the water contact angle and significantly improved the wettability of the membrane surface.

Water stained by methylene blue was placed onto the surface of ODA-6FDA and IL-ODA-6FDA/PVP@PDA membranes in drops. As shown in Figure 6b, IL-ODA-6FDA/PVP@PDA membranes can quickly absorb droplets dyed with the dimethyl blue, taking less than 5 s to fully do so. However, the droplets stayed on the original ODA-6FDA surface in their entirety as per Figure 6a.

In Young’s Equation (DF = G ± Pc) [30], the terms DF, G, and Pc represent, the driving force, gravity, and additional pressure at the gas-liquid interface of a spherical water droplet (Figure 7), respectively. Specifically, Pc = 4γcos θ/d, within which γ is the surface tension of water, θ is the water contact angle, and d is the pore size. It is assumed the direction of gravity is positive. Due to the hydrophobic nature of the ODA-6FDA membrane, the unmodified membrane had a contact angle of 126–129° (i.e., greater than 90°) (Figure 5), and because cos θ < 0 and Pc < 0, Pc lay in the opposite direction to G (Figure 7a), and so DF = G − Pc is applied, resulting in droplets remaining on the membrane surface. Conversely, for the IL-ODA-6FDA/PVP@PDA membrane, its hydrophilic nature resulted in θ = 10–28° (that is, less than 90°) (Figure 5), cos θ > 0, and Pc > 0, therefore necessitating the use of the alternate expression DF = G + Pc, enabling absorption of droplets on the membrane surface. As mentioned above, IL-ODA-6FDA/PVP@PDA membranes prepared by electrospinning showed excellent hydrophilicity, which raises its potential to become an ideal material for processes involved in sewage treatment in the future.

### 3.5. Oil-Water Separation Performance of Membrane

Mixtures of water with non-emulsified petroleum ether, gasoline, n-hexane, or toluene were individually prepared. With each of these, the separation performance of the membranes was tested. To begin with, the IL-ODA-6FDA/PVP@PDA membrane was wetted with deionized water, then, in separate tests, each of the above oil/water mixtures was poured into the suction filter bottle where the porous membrane had been placed. Figure 8 illustrates the separation of the petroleum ether/water mixture by IL-ODA-6FDA/PVP@PDA membrane. Under the condition of relying on their own weight, 15 mL of each oil-water mixture was completely separated within 30 min. All the water dyed by methylene blue passed quickly, while the oil was almost completely intercepted. The results suggested the IL-ODA-6FDA/PVP@PDA fiber membrane had a better separation effect than unmodified variants on the oil-water mixture.

The oil/water emulsions were observed under magnification with an optical microscope. From Figure 9, oil droplets surrounded by water in emulsion were very recognizable. After the emulsions were separated by the IL-ODA-6FDA/PVP@PDA membrane, filtrates became transparent, showing no signs of oil-impregnated emulsion. Nanometer particle size analysis characterized and compared the average particle size. Before separation, the average particle size of the prepared emulsified oil was mainly in the range of 712.75 ± 3.14 nm (Figure 10a), while after it was reduced to 27–37 nm (Figure 10c). This indicates the IL-ODA-6FDA membrane can block hydrophobic content and, based on the distribution range, separates petroleum ether–water emulsion systems well. As demonstrated in Figure 10e, the average size of the emulsion droplets after separation by IL-ODA-6FDA/PVP@PDA, was recorded to be in the range of 10–13.5 nm. Due to the addition of a large number of hydrophilic groups from IL and PVP, and the influence of PDA on the membrane surface roughness of the film, the particle size data showed that this membrane variant has excellent separation ability when applied to emulsions.

For the emulsified oil-water separation performance test, the IL-ODA-6FDA/PVP@PDA membrane was assessed using four different oil-water emulsions (with toluene, n-hexane, petroleum ether, and chloroform as the hydrophobic components). For the preparation of the emulsions, their oil content before separation was 10%, each sample was of diameter 1.5 cm, the test time was 15 min, and there was a total of 20 samples. On the basis of it being conducted as a non-circulating test, each sample was tested five times. The separation test of the emulsified oils followed the same steps as that of the non-emulsified oil-water systems, i.e., sieved by the weight of the liquid. This process thus used the same apparatus as illustrated in Figure 7, with the separation effects as shown in Figure 8. The membrane flux of ODA-6FDA for the different emulsified oil of systems was maintained at 120–130 L·m^−2^·h^−1^, while the separation efficiency fell within the range of 23–36% (Figure 11a). By contrast, these respective properties of IL-ODA-6FDA were 170–200 L·m^−2^·h^−1^ and 70%. Therefore, the membrane separation efficiency was improved due to the introduction of an ionic liquid component containing short carbon chains, whose high surface energy allows water droplets to diffuse and penetrate the surface quickly.

IL-ODA-6FDA/PVP has a significantly improved in membrane surface wettability over the variant not treated with PVP. Its membrane flux was maintained at 180–240 L·m^−2^·h^−1^, with a separation efficiency which remained above 85%. When further modified by PDA, the membrane displayed increases in both membrane flux and separation efficiency. Over the whole separation process, the maximum flux obtained by using IL-ODA-6FDA/PVP@PDA membrane for N-hexane/water emulsion exceeded 290 L·m^−2^·h^−1^, and the maximum separation efficiency reached was more than 99%. In addition, the IL-PI/PVP@PDA membrane showed similar performance (also higher than 99% separation efficiency) across all types of emulsion it was tested against (Figure 11d). The self-polymerization of PDA on the IL-ODA-6FDA/PVP fiber surface contributed to this by providing a large number of hydrophilic groups, such as -NH and -OH. The addition of these hydrophilic groups decreased θ, meaning the positiveness of Pc increased. In turn, this ultimately led to an increase in the driving force as evidenced by the water droplets on the IL-ODA-6FDA/PVP@PDA film completely wetting the membrane and spreading across the film surface upon contact. The micro and nano structure of the fiber membrane itself allowed the screening of some emulsified oils by large particle sizes. In Figure 2d, as the membrane’s hole spacing decreases, Pc becomes larger, and so DF also increases, promoting overall separation efficiency. The intermolecular forces between the hydrogen bonding sites provided by the surface and the receptor are able to separate the components of the emulsions by increasing the contact area with water.

### 3.6. Recyclability of IL-PI/PVP@PDA Fibrous Membrane

Recyclability is another important property for which a technology must be held to a high standard for future application of porous membranes in oil-water separation. For this, the cycle performance of the membrane was investigated. The separation efficiency of all porous membranes only changed slightly after 10 consecutive cycles of petroleum ether/water emulsion (Figure 12). That of IL-ODA-6FDA/PVP@PDA membrane did not fall below 95% after these iterative tests, suggesting a high level of stability and reusability.

At present, industrial requirements for oil-water separating membranes must not only feature proven, high separation efficiency, but also the capacity for large-scale production, and excellent mechanical properties and cycle capabilities. The porous membranes prepared as detailed here introduce a large number of hydrophilic groups [33], while taking into account the excellent mechanical strength of PI itself. The separation flux was demonstrated to reach 290 L·m^−2^·h^−1^, with a separation efficiency exceeding 99%. Compared with other experimental membranes [33,34], the straightforward preparation of IL-ODA-6FDA/PVP@PDA and its positive test results ensure its stable water flux and high separation efficiency. This also means it meets the above requirements for separation membranes with potential industrial applications, thus affording it realistic commercial prospects.

## 4. Conclusions

By introducing the ionic liquid IL and PVP, the modified PI membrane prepared by electrospinning had significantly enhanced mechanical properties compared with the pure unmodified membrane. To develop further on these findings, in-situ growth of PDA was used to improve the membrane’s wettability. This addition caused the membrane’s water contact angle to fall from the IL-PI/PVP membrane’s 128.97 ± 3.86° to 30.26 ± 2.16°. Fully modified, through testing on various oil-water emulsions, its separation performance was demonstrated to be exemplary, attaining a water flux of 290 L·m^−2^·h^−1^ and separation efficiency 99% with no external force acting on the emulsion other than gravity. Even after 10 successive test cycles, separation efficiency remained greater than 95%. As proven by the data presented here, this method of porous membrane modification will provide a foundation for future technologies and their practical applications, particularly in wastewater treatment.

## Figures and Tables

**Figure 1 membranes-12-00189-f001:**
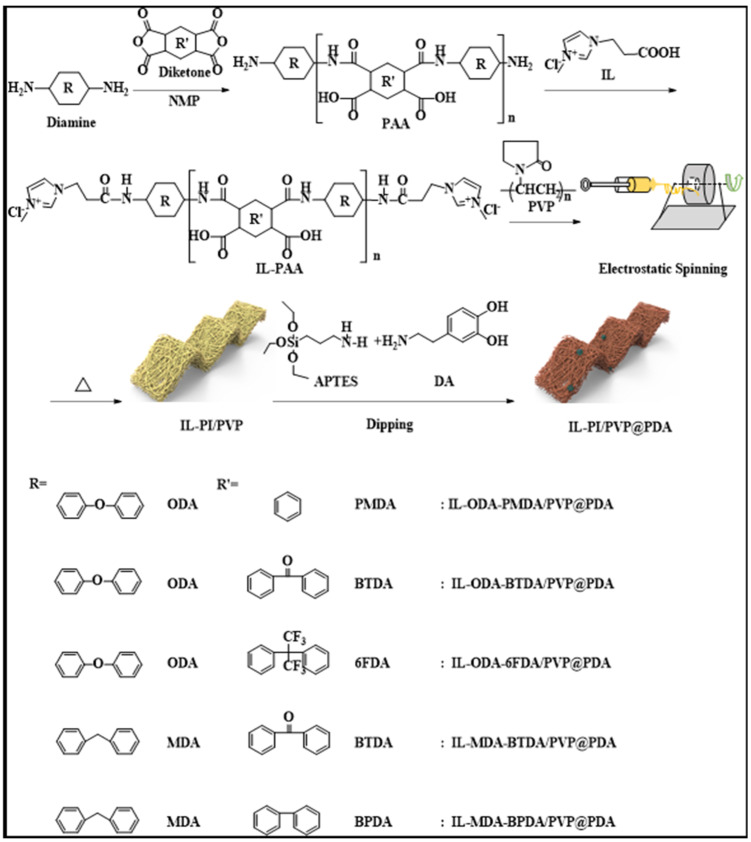
The preparation process of IL-PI/PVP@PDA porous membranes.

**Figure 2 membranes-12-00189-f002:**
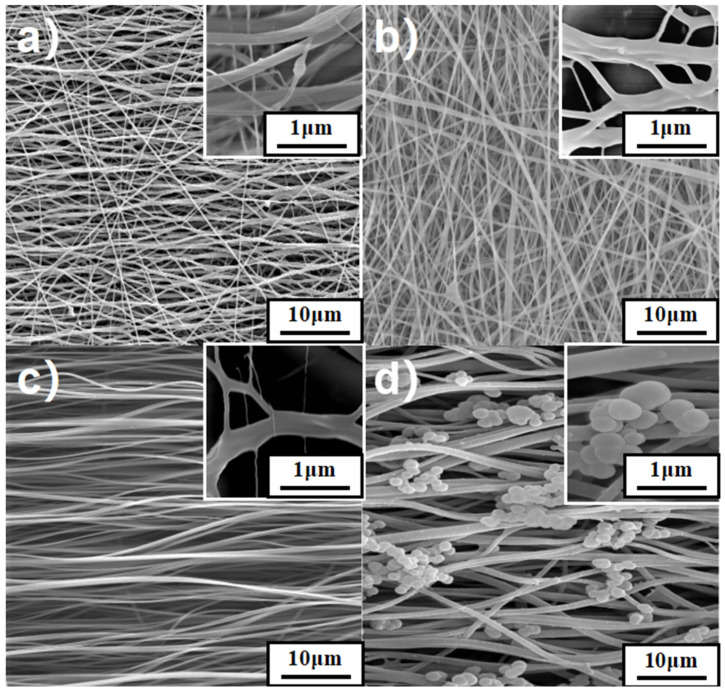
SEM images of electrostatic porous membrane: (**a**) ODA-6FDA, (**b**) IL-ODA-6FDA, (**c**) IL-ODA-6FDA/PVP, (**d**) IL-ODA-6FDA/PVP@PDA in Table 1.

**Figure 3 membranes-12-00189-f003:**
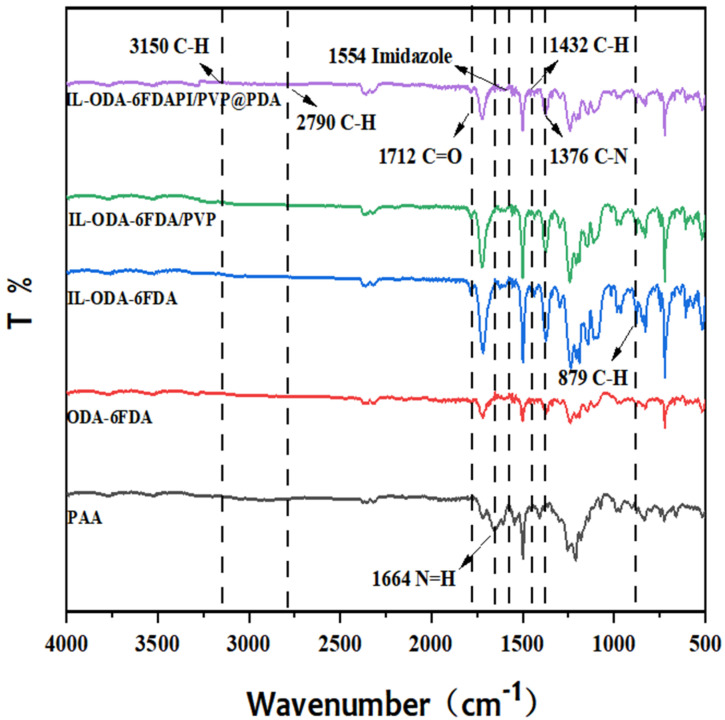
FT-IR spectra of electrostatically porous membrane.

**Figure 4 membranes-12-00189-f004:**
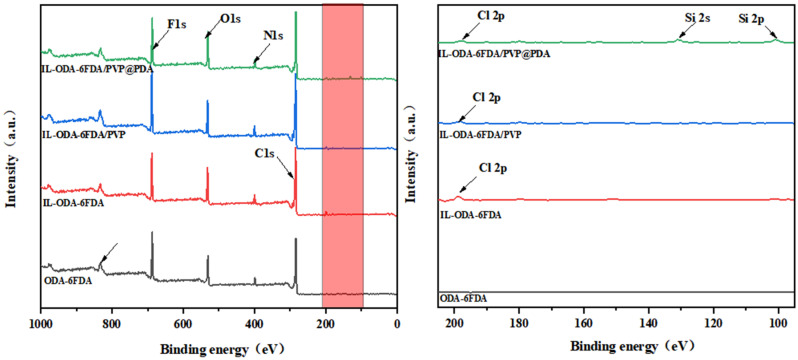
XPS spectra of PI electrostatically porous membrane.

**Figure 5 membranes-12-00189-f005:**
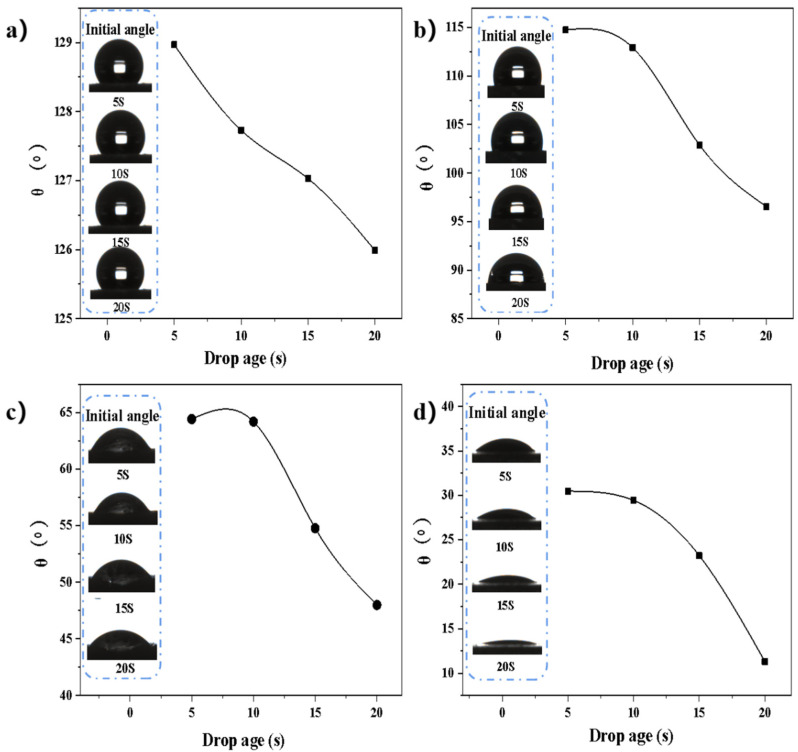
The water contact angle (θ) of the PI porous membranes at different times: (**a**) ODA-6FDA, (**b**) IL-ODA-6FDA, (**c**) IL-ODA-6FDA/PVP, (**d**) IL-ODA-6FDA/PVP@PDA in Table 2.

**Figure 6 membranes-12-00189-f006:**
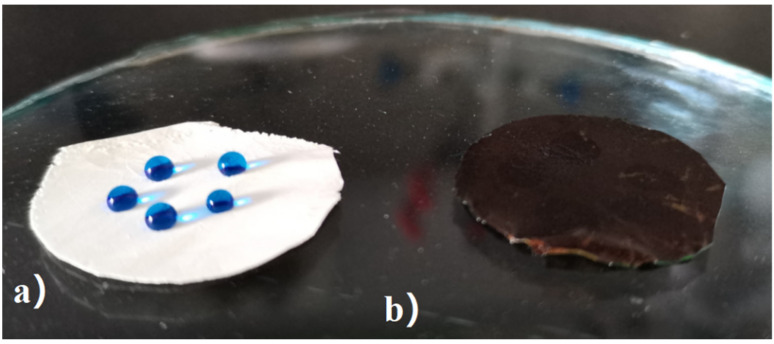
Photographs displaying droplets on the film surface (**a**) original ODA-6FDA in air, (**b**) IL-ODA-6FDA/PVP@PDA membrane in Table 2.

**Figure 7 membranes-12-00189-f007:**
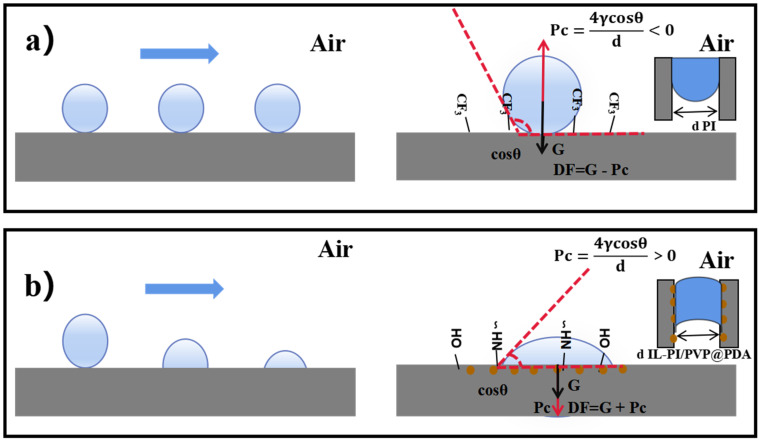
Schematic diagram of the separation principle of original and modified ODA-6FDA membranes, (**a**) water on the surface of the unmodified membrane in air, and (**b**) water on the surface of IL-PI/PVP@PDA membrane in air.

**Figure 8 membranes-12-00189-f008:**
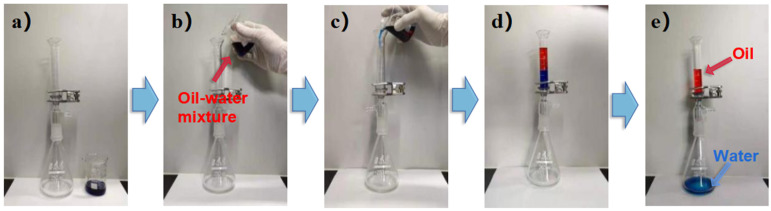
The separation process of IL-ODA-6FDA/PVP@PDA porous membrane. (**a**) get the device ready; (**b**) and (**c**) pour the oil-water mixture into the filter bottle; (**d**) oil-water mixture separation process; (**e**) oil and water can be completely separated by their own weight.

**Figure 9 membranes-12-00189-f009:**
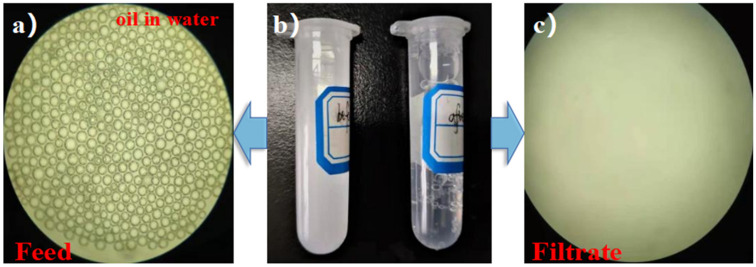
Optical microscope images of petroleum ether/water emulsion, (**a**) before separation, (**b**) macro photos of the emulsion before and after separation, (**c**) after separation.

**Figure 10 membranes-12-00189-f010:**
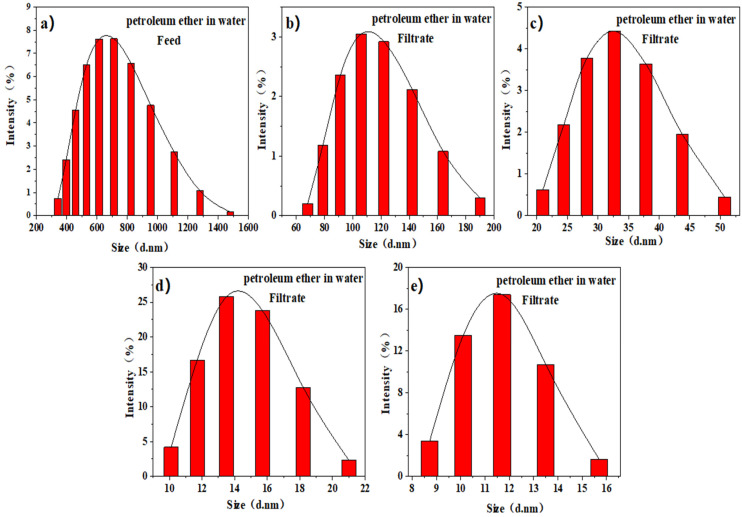
Nanoparticle size analysis of petroleum ether/water emulsion before and after PI membrane separation at different stages (**a**) before separation, (**b**) ODA-6FDA, (**c**) IL-ODA-6FDA, (**d**) IL-ODA-6FDA/PVP, and (**e**) IL-ODA-6FDA/PVP@PDA membrane in Table 2.

**Figure 11 membranes-12-00189-f011:**
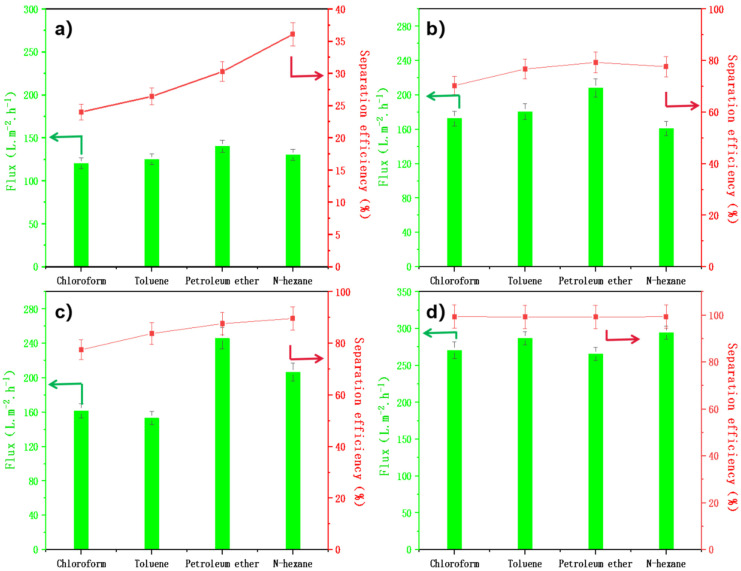
Porous membrane separation performance test for emulsified oil water mixture, (**a**) ODA-6FDA, (**b**) IL-ODA-6FDA, (**c**) IL-ODA-6FDA/PVP, and (**d**) IL-ODA-6FDA/PVP@PDA in Table 2.

**Figure 12 membranes-12-00189-f012:**
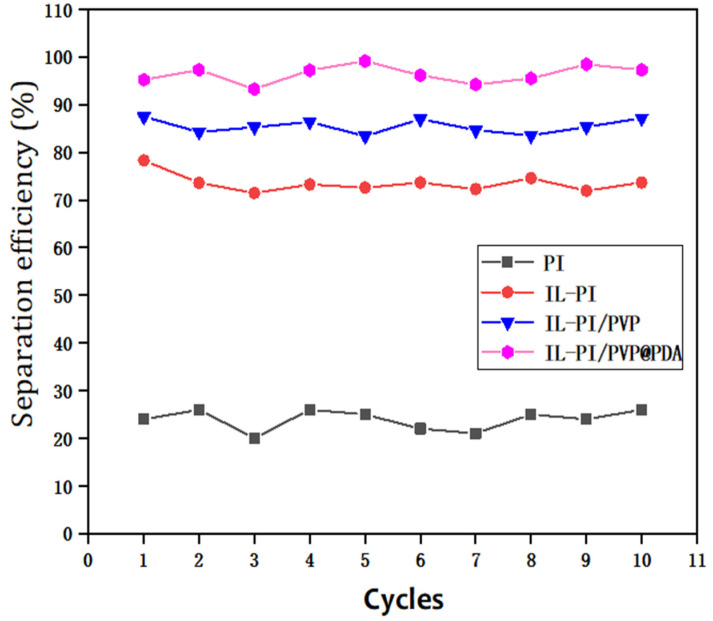
Recycling performance test of the porous membranes.

**Table 1 membranes-12-00189-t001:** Porosity of modified PI membranes.

Membrane	Basis Weight a (g/m^2^)	Pore Size b (μm)	Porosity c (%)
ODA-6FDA	25.72	3.53	7.09
IL-ODA-6FDA	22.67	4.62	17.48
IL-ODA-6FDA/PVP	28.94	4.89	19.02
IL-ODA-6FDA/PVP@PDA	20.11	3.95	12.76

Notes: (a) Measurement with an analytical balance, (b) membrane pore size measurement (direct method), (c) measurement of porosity (weighing method).

**Table 2 membranes-12-00189-t002:** Mechanical and oil-water separation properties of PI and IL-PI membranes ^a^.

No.	Designation	Εb ^b^ (%)	E ^c^ (MPa)	F ^d^ (L·m^−2^·h^−1^)	R ^e^ (%)	θ ^f^ (°)
1	ODA-6FDA	8.90 ± 0.14	459.12 ± 0.22	120.27 ± 1.20	24.01 ± 6.01	128.97 ± 3.86
2	IL-ODA-6FDA	26.95 ± 0.44	2271.23 ± 1.01	172.45 ± 3.51	70.27 ± 8.62	114.70 ± 2.17
3	IL-ODA-6FDA/PVP	39.21 ± 0.31	3056.46 ± 1.26	161.31 ± 8.06	77.45 ± 3.87	64.43 ± 1.26
4	IL-ODA-6FDA/PVP@PDA	37.74 ± 0.47	3198.38 ± 0.78	270.40 ± 11.52	99.31 ± 0.16	30.26 ± 2.16
5	MDA-BTDA	20.50 ± 0.12	314.50 ± 0.09	139.63 ± 5.62	32.52 ± 7.92	103.50 ± 2.19
6	IL-MDA-BTDA	21.27 ± 0.36	1573.12 ± 1.30	150.74 ± 6.42	68.42 ± 5.63	87.65 ± 1.12
7	IL-MDA-BTDA/PVP	32.62 ± 0.52	2443.79 ± 1.26	190.59 ± 8.29	78.62 ± 7.94	54.73 ± 0.42
8	IL-MDA-BTDA/PVP@PDA	39.70 ± 0.43	3698.73 ± 0.96	204.74 ± 7.94	99.57 ± 0.23	29.02 ± 0.46
9	ODA-PMDA	2.84 ± 0.05	729.12 ± 0.17	148.74 ± 10.02	24.95 ± 6.96	74.10 ± 0.23
10	IL-ODA-PMDA	6.20 ± 0.03	3057.40 ± 0.70	136.79 ± 6.35	70.62 ± 5.53	56.85. ± 0.85
11	IL-ODA-PMDA/PVP	12.97 ± 0.32	3131.78 ± 0.16	176.83 ± 4.23	83.22 ± 3.97	54.23 ± 1.84
12	IL-ODA-PMDA/PVP@PDA	19.52 ± 0.02	3023.10 ± 1.24	286.41 ± 8.49	98.68 ± 0.21	18.16 ± 0.58
13	MDA-BPDA	14.35 ± 0.52	496.42 ± 1.58	105.83 ± 9.73	32.86 ± 8.84	136.43 ± 0.78
14	IL-MDA-BPDA	25.40 ± 0.07	1950.73 ± 1.91	164.72 ± 12.84	65.85 ± 6.29	82.52. ± 0.31
15	IL-MDA-BPDA/PVP	36.27 ± 0.06	2122.99 ± 1.06	210.58 ± 14.72	73.91 ± 3.42	64.40 ± 2.46
16	IL-MDA-BPDA/PVP@PDA	34.62 ± 0.26	2669.10 ± 1.84	283.82 ± 9.64	94.98 ± 1.83	21.11 ± 0.74
17	ODA-BTDA	2.20 ± 0.12	391.46 ± 0.09	135.85 ± 13.57	37.07 ± 5.97	96.30 ± 1.35
18	IL-ODA-BTDA	11.28 ± 0.03	2515.12 ± 1.30	182.97 ± 11.46	57.33 ± 8.31	76.49 ± 0.84
19	IL-ODA-BTDA/PVP	19.90 ± 0.05	2269.40 ± 0.17	205.63 ± 9.64	72.86 ± 4.31	69.64 ± 1.48
20	IL-ODA-BTDA/PVP@PDA	31.53 ± 0.03	3797.34 ± 0.70	263.24 ± 8.32	96.43 ± 1.97	15.31 ± 0.72

Notes: ^a^ The standard spline with a length of 50 mm, a width of 20 mm, and a test speed of 5.00 mm/min, ^b^ elongation at break (εb), ^c^ modulus of elasticity, ^d^ film flux, ^e^ separation efficiency, and ^f^ θ (the water contact angle).

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
