# Peer review of "Preparation and Separation Properties of Electrospinning Modified Membrane with Ionic Liquid Terminating Polyimide/Polyvinylpyrrolidone@Polydopamine"

_membranes, 2022, doi:10.3390/membranes12020189_

Round 1

Reviewer 1 Report

Dear Authors,

The manuscript describes nonwoven mats of a polymer composite containing ionic liquid capped main chain-polymers mixed with polyvynylpirrolidone as oil-water separation membranes as potential sewage treatment technologies.

Concerning the results, the size caracterization of the filtrates nanoparticles is niether described on the experimental part and not enough discussed in the results. Is it performed with a Z-sizer or DLS?

SEM images justifying the aggregates of the PDA can lead to misinterpreted results, if a phase separation occurs, the overall performance of the membrane can lead to false positive results due to preferential ways of the mean average pathway because the surface functionalization is not homogeneous. 

The PVP and the chemical reactions are not accurate (it is a pirrolidone, not a pirimidine). 

The siloxane functionalization with APTES is not accurate neither, the silane has three substitutional sites. Might this observation could suggest a partial SiO2 cluster formation on the fibers due to silanol polymerizations? Please explain, justify and rephrase. 

The results are not compared with references according with performance.  

 Based on the above observations I regret to inform you that the manuscript cannot be accepted in its present form. 

Author Response

Dear Editor:

Thank you very much for reviewing our manuscript “Electrostatic spinning fiber membrane of polydopamine modifying and ionic liquid terminating polyimide/polyvinylpyrrolidone and its oil-water separation performance” to be considered for publication in “Membranes”.

”.

We have revised our manuscript considering all the reviewers’ comments. The revisions are shown by Track Changes function in the revised manuscript. The point-to-point replies to three reviewers follow to this cover letter. We believe that the manuscript has now been revised enough to be suitable for publication in “Membranes”.

Please let us know your decision at your earliest convenience.

                                                                                 Thank you and best regards.

                                                                                                Hongge Jia

Reviewer 2 Report

طز

The manuscript investigated ” Electrostatic spinning fiber membrane of polydopamine modi- 2 fying and ionic liquid terminating polyimide/polyvinylpyrroli- 3 done and its oil-water separation performance”. The findings in the paper are of significance in engineering application to Evaluating the Electrostatic spinning fiber membrane of polydopamine. It is a meaningful work, but I think the manuscript still has a lot of problems and needs to do more work to improve the quality of this manuscript to meet the standards for publication in Journal of Membranes. Under the present state, I therefore suggest a Major revision of the manuscript. I have added a detailed list of comments below: they should be taken into account by the authors when reworking the manuscript.

Author Response

(The authors gave the same response as above.)

Reviewer 3 Report

The manuscript reported the preparation of hydrophilic nanofiber membrane by the electrospinning approach and different treatments. Superhydrophobic polyimide (PI) was selected as the basic supporter of membrane and serval treatments are applied to improve the hydrophilic property. Then the filtration and separation of the oil-water emulsion experiment were applied to evaluate the property of different membranes.

I consider the content of this manuscript will meet the reading interests of the readers of the Membranes journal. However, there are certain English spelling and grammar issues, and the meaning of this work needs to be emphasized, as well as the results and discussion should be further improved.

Therefore, I suggest giving a major revision and the authors need to clarify some issues or supply some more experimental data to enrich the content. This could be a comprehensive work after revision.

Author Response

(The authors gave the same response as above.)

Round 2

Reviewer 1 Report

Dear authors,

The manuscript of the present research has been considerably improved, although, it is necessary to check again the next special observations:

On the page 2 line 81: it is suggested to change "good" for "suitable" to avoid ambiguity

Page 3, line 97:  "reagents" or "chemicals" shall be written instead of "drugs". 

Page 4, lines 112-118: A more technical vocabulary shall be employed to describe the device configuration (e.g. the rotating cylinder it is called the collector)

Page 4 line 116: Concerning the 6FDA addition, The number of batches prepared shall be specified. 

Concerning the thermal treatment of the nonwoven mats, the heating ramp, and the dwelling time of the curing temperature shall be written down instead the word "gradually". (e.g. 10 ºC/ min until reaching 280 ºC for x h) 

It is required to specify the chemical reaction/weak interaction between APTES and the polymer functional groups, otherwise the formation of your silica layer can be due to a formation of silica particles deposited within the porosity of the mat.

Concerning the XRF device, I am pretty curious if the model is the D8, because this code from bruker (as far as I know, and as checked at their website) corresponds to a diffractometer. I believe that the proximity with the "s" key led to a typing mistake. Please check.

Based on the previous comments, the article can be accepted if the queries are answered. 

Reviewer 2 Report

Accepted

Author Response

  Thank you very much for your reply

翻译结Thank you very much for yo

Reviewer 3 Report

I have carefully read the author's reply to the reviewer. I consider the author fully respected all the previous suggestions and comments proposed by me, and have made serious point-to-point replies and considerable revisions within the scope of the full text. It can be seen that the author worked very hard in the revision process.

I can no longer put forward any other suggestions and comments to improve this manuscript.

Author Response

Thank you very much for your reply